# Multiple Sclerosis, Disease-Modifying Therapies and COVID-19: A Systematic Review on Immune Response and Vaccination Recommendations

**DOI:** 10.3390/vaccines9070773

**Published:** 2021-07-11

**Authors:** Verónica Cabreira, Pedro Abreu, Ricardo Soares-dos-Reis, Joana Guimarães, Maria José Sá

**Affiliations:** 1Serviço de Neurologia, Centro Hospitalar Universitário de São João, 4200-319 Porto, Portugal; pedro.abreu@chsj.min-saude.pt (P.A.); ricardo.soares.reis@chsj.min-saude.pt (R.S.-d.-R.); joana.guimaraes@chsj.min-saude.pt (J.G.); mjsa@med.up.pt (M.J.S.); 2Departamento de Neurociências Clínicas e Saúde Mental, Faculdade de Medicina da Universidade do Porto, 4200-319 Porto, Portugal; 3i3S-Instituto de Investigação e Inovação da Universidade do Porto, 4200-135 Porto, Portugal; 4Faculdade de Ciências da Saúde, Universidade Fernando Pessoa, 4249-004 Porto, Portugal

**Keywords:** COVID-19, vaccines, multiple sclerosis, immunosuppression, multiple sclerosis/therapy

## Abstract

Understanding the risks of COVID-19 in patients with Multiple Sclerosis (MS) receiving disease-modifying therapies (DMTs) and their immune reactions is vital to analyze vaccine response dynamics. A systematic review on COVID-19 course and outcomes in patients receiving different DMTs was conducted according to the Preferred Reporting Items for Systematic Reviews and Meta-Analyses statement. Emerging data on SARS-CoV-2 vaccines was used to elaborate recommendations. Data from 4417 patients suggest that MS per se do not portend a higher risk of severe COVID-19. As for the general population, advanced age, comorbidities, and higher disability significantly impact COVID-19 outcomes. Most DMTs have a negligible influence on COVID-19 incidence and outcome, while for those causing severe lymphopenia and hypogammaglobulinemia, such as anti-CD20 therapies, there might be a tendency of increased hospitalization, worse outcomes and a higher risk of re-infection. Blunted immune responses have been reported for many DMTs, with vaccination implications. Clinical evidence does not support an increased risk of MS relapse or vaccination failure, but vaccination timing needs to be individually tailored. For cladribine and alemtuzumab, it is recommended to wait 3–6 months after the last cycle until vaccination. For the general anti-CD20 therapies, vaccination must be deferred toward the end of the cycle and the next dose administered at least 4–6 weeks after completing vaccination. Serological status after vaccination is highly encouraged. Growing clinical evidence and continuous surveillance are extremely important to continue guiding future treatment strategies and vaccination protocols.

## 1. Introduction

Coronavirus disease 2019 (COVID-19) is a severe acute respiratory syndrome caused by a variant of coronavirus (SARS-CoV-2). It emerged in December 2019, in the Wuhan province and rapidly spread worldwide [1]. After SARS-CoV-2 infection, the innate immune system (macrophages, dendritic cells, natural killer (NK) cells) triggers signalizing cascades leading to an increase of pro-inflammatory cytokines. Simultaneously, significant T lymphocyte subside, limiting the establishment of adaptative immunity and the host ability to solve the infection [2]. Later, to establish a prolonged immunity and reduce the risk of re-infection, plasmacytes and memory B and T cells take place to produce neutralizing antibodies. Since the outbreak of the pandemic, in addition to the well-known COVID-19 symptoms, including neurological ones [1], there has been a concern with populations facing a higher risk of infection [3]. Multiple sclerosis (MS) patients are usually under disease modifying therapies (DMTs) and have been identified as a high-risk population, both because inflammation is an integral part of MS pathogenesis, and infections are well known exacerbation triggers of MS-related disease activity [4].

While many immunotherapies, including DMTs used for MS, have been pointed as COVID-19 candidate therapies [5], the development of SARS-CoV-2 vaccines was early assumed to be the turning point for the still expanding pandemic, and a high priority research topic. Now that many vaccines have been developed and approved by the regulatory agencies, the vaccination campaigns are speeding up all over the world. Notwithstanding, besides the susceptibility and clearance of SARS-CoV-2 infection in patients under different DMTs, the question is now turning to understanding the vaccine response in MS patients. This will be the key to define the best vaccination strategy and a guarantee of effective immune response after vaccination.

## 2. Materials and Methods

We conducted a systematic review on the incidence and outcomes of COVID-19 in MS patients on DMTs, according to the Preferred Reporting Items for Systematic Reviews and Meta-Analyses (PRISMA) statement [6]. The following query was introduced to PubMed database: multiple sclerosis AND (SARS-CoV-2 OR COVID-19) AND (disease-modifying therap* OR DMT OR interferon OR glatiramer acetate OR teriflunomide OR dimethyl fumarate OR fumarate OR fingolimod OR natalizumab OR ocrelizumab OR rituximab OR cladribine OR alemtuzumab). A clinical trial database and conference abstracts were also analyzed. After selecting the articles related to the first part of this review, we looked for articles on ‘Multiple Sclerosis’; ‘COVID-19’; ‘SARS-CoV-2’; ‘vaccines’ and ‘vaccination’ to analyze the immune response dynamics to vaccines on MS patients. We focused on the approved SARS-CoV-2 vaccines and the data available on their safety and efficacy, including some recent data in patients with MS. Experience with previous vaccines in relation with DMTs was also deemed pertinent as a window for the new vaccines. Using this current evidence, the second part of the document was designed to elaborate expert recommendations that might guide vaccination protocols. Two qualified authors evaluated independently the titles and abstracts of each article and decided to include only original articles, case series (including safety data registries), case reports, letters, correspondence and short communications. Systematic reviews were not considered. Full text of included articles was reviewed for the extraction of patients’ clinical features. When we identified duplicated results, those articles were removed.

## 3. Results

Study selection according to PRISMA criteria is represented in a flow-chart (Figure 1). A total of 164 articles were retrieved from PubMed (last search on 27 April) as well as two conference abstracts. Of these, 80 articles were included in the review. Reports on vaccination and MS, as well as papers recovered from reference lists, were added, totalizing 110 studies meeting the final inclusion criteria.

Several studies have already reported on SARS-CoV-2 infection and prognosis in MS patients under different DMTs. So far, 4417 MS patients were reported to have been infected with SARS-CoV-2, with 99 reported dead due to COVID-19 complications (Table 1).

Since lymphopenia is a rare event for patients under interferons-beta (IFN-beta), this treatment is unlikely to interfere with susceptibility and immune response to SARS-CoV-2. We identified a total of 218 MS patients on IFN treatment with a confirmed (*n* = 144) or suspected (*n* = 74) COVID-19 diagnosis [7,8,9,10,11,12,13,14,15,16,17,18,19,20,21,22,23,24,25]. Four (1.8%) of them were reported dead due to COVID-19 complications [21,25]. In the Italian real-world dataset, among the 73 patients on IFN, no ICU admissions or deaths were recorded. IFN-beta has been associated with the lowest risk of SARS-CoV-2 infection across all available series, comparing to other DMTs [18]. A mild protective effect of IFN-beta might possibly be related to the reversal of SARS-CoV-2 inhibition of type 1 IFN molecules production and intracellular antioxidant nuclear factor erythroid 2–related factor 2 pathway [9,26,27], which is further emphasized by reports of severe cases of COVID-19 among patients with genetic and non-genetic deficiencies in interferon system components, such as low expression of IFN receptor gene or anti-IFN neutralizing antibodies [28]. However, available clinical trial data suggest that IFNs (used alone or in combination with other antiviral agents) have yielded mixed results in patients with SARS-CoV-2.

Glatiramer acetate (GA) competes with myelin antigens for the interaction with histocompatibility major complex molecules in antigen-presenting cells, shifting from a pro-inflammatory to an anti-inflammatory response. As IFN, it does not promote lymphopenia. Furthermore, GA blocks IFN gamma mediated activation of macrophages, which is thought to play an essential role in acute respiratory distress syndromes [10,26]. We found a total of 269 patients under GA with confirmed (*n* = 196) or suspected (*n* = 73) infection by SARS-CoV-2 [8,10,11,15,16,18,19,20,21,22,23,24,25,29,30,31]. Four (1.5%) of these patients died due to COVID-19 complications, one of them with a secondary progressive MS (SPMS) and several comorbidities (past medical history of venous thromboembolism and obesity) [11,18,25]. In general, there is no evidence of an increased infectious risk during treatment with IFN or GA, and both are safe medications for MS patients during the SARS-CoV-2 pandemic. Even so, whether the reduced risk during IFN and GA therapy is driven by immune modulation or by patient and disease characteristics (usually milder disease activity and less disability) remains to be determined.

Teriflunomide selectively and reversibly inhibits dihydro-orotate dehydrogenase, an essential mitochondrial enzyme in the de novo pyrimidine synthesis pathway [32,33]. The drug has a dual antiviral and immunomodulatory action: it decreases the proliferation of reactive lymphocytes, preventing fulminant host immune responses, and halts the viral replication inside the infected cell. Past studies have suggested a possible effect of teriflunomide against several viruses, including respiratory syncytial virus, ebola, cytomegalovirus, Epstein–Barr and variants of picornavirus [26,27,32], at expense of a slight increase in the risk of viral upper respiratory tract infections, possibly due to secondary lymphopenia (12%) and neutropenia (16%) [26]. We identified a total of 251 patients with confirmed (*n* = 202) or suspected (*n* = 49) diagnosis of COVID-19 [7,8,10,13,15,16,17,18,19,20,21,22,23,24,25,31,32,33,34,35,36,37,38,39], all with mild and self-limiting courses except for four (1.6%) patients who had a fatal outcome (at least one of the patients had a SPMS and an Expanded Disability Severity Scale (EDSS) of 7.5 plus myotonic dystrophy) [10,25]. For one teriflunomide-treated MS patient, the COVID-19 presentation was dominated by visual symptoms that completely subsided, without the need to interrupt the drug [39]. The reported cases presented with a variable recovery time, ranging from a few days to some weeks. For the few patients who required hospitalization and for whom teriflunomide had to be stopped, therapy was started soon after hospital discharge with no intercurrences. Further, an analysis of peripheral blood immune cell profile in a teriflunomide-treated MS patient found normal immune T cell activation before and during SARS-COV-2 infection [33]. Considering the evidence, it seems safe to carry on with teriflunomide treatment during COVID-19 pandemic, even throughout the acute infection period.

Regarding Dimethyl fumarate (DMF), its immunomodulatory mechanism seems to depend on Nfr-2 protein inhibition, limiting inflammatory cascades including macrophage activation. Perhaps the most important factor regarding SARS-CoV-2 infection is the fact that lymphopenia is a well-recognized idiosyncratic effect with this drug. It occurs in at least 37% of the patients (severe <500/mm^3^ in 8%) and particularly affects the CD8 T cells and memory cells. We found 603 patients with confirmed (*n* = 408) or suspected (*n* = 195) COVID-19 while on DMF [8,11,12,13,14,15,16,17,18,19,20,21,22,23,24,25,29,31,35,40,41]. Nineteen (3.2%) patients are reported dead due to COVID-19 complications [18,21,25]. One of them was a 68-year-old man with SPMS, EDSS of 6.0 and a past medical history of cerebrovascular disease and hypertension, both of which are known negative prognostic factors for COVID-19 [18]. Despite the majority had mild and self-limiting courses, lymphocyte count should be carefully monitored in patients treated with DMF, and treatment suspended for lymphocyte count under 800/mm^3^, given the higher risk of infection in these patients. It appears safe to proceed with the therapy in the absence of lymphopenia and for asymptomatic or mildly symptomatic patients.

Fingolimod is a sphingosine-1-phosphate (S1P) receptor modulator that prevents lymphocytes from leaving the lymph nodes, inhibiting autoimmune reactions. By sequestering lymphocytes, instead of promoting direct lymphocyte exhaustion, it reduces the total mean circulating lymphocyte count, which usually remains above 200/mm^3^ [26,42]. Nonetheless, it may increase the risk of mild infections, mainly viral infections, including influenza, herpes virus, JC virus and possibly SARS-CoV-2 infection. On the other hand, a potential beneficial effect in preventing reactive «cytokine storm» and enhancing lung endothelial cell integrity has been discussed. An exploratory study to evaluate the efficacy of fingolimod for COVID-19 was initiated (https://clinicaltrials.gov/ct2/show/NCT04280588, accessed on 27 April 2021) but recruitment was early interrupted. We found 414 COVID-19 cases under fingolimod (confirmed *n* = 268, suspected *n* = 146) [7,8,10,11,14,15,16,17,18,19,20,22,23,24,25,30,31,35,37,43,44,45,46,47,48,49]. One of these patients, a 42-year-old female with relapsing-remitting MS (RRMS), an EDSS of 6.0 and some comorbidities (severe cognitive impairment and struma ovarii treated with radioiodine), who refused ICU admission, died [20]. Nineteen cases of COVID-19 under siponimod and one suspected COVID-19 case under ponesimod have been described in the literature [11,18,25]. Further 29 cases under S1P modulators (non-specified) were described [21]. Two patients died due to COVID-19 while on siponimod [25]. Overall, most patients had a relatively benign disease course, despite negative prognostic features and lymphopenia, with prompt recovery, even if ICU was initially needed. To note, cases of clinical exacerbation of SARS-CoV-2 infection after fingolimod withdrawal have been described. A 57-year-old male with RRMS and EDSS 6.0 showed hyper-inflammation syndrome one week after fingolimod withdrawal, with progressive improvement after steroid therapy [47]. So far, the use of fingolimod does not seem to expose people to a particular risk of unfavorable COVID-19 evolution and the risk of aggressive rebound of MS activity with fingolimod’s discontinuation [44,47] probably outweighs the risk of infection, contradicting treatment discontinuation. As so, it appears safe to start or maintain fingolimod treatment, including during SARS-CoV-2 acute infection period, with the need to withdrawal therapy only if lymphocyte count falls under 200/mm^3^.

Natalizumab is a humanized monoclonal antibody against α4-integrin that acts by preventing the lymphocyte migration to the central nervous system (CNS). Some studies advanced that SARS-CoV-2 may as well use integrin to enter the human cells and so a hypothetical beneficial protective effect of natalizumab was suggested. We identified 461 COVID-19 cases (confirmed *n* = 325, suspected = 136) [8,11,13,14,15,16,18,19,20,21,22,23,24,25,30,31,35,50,51,52]. Many of these patients were asymptomatic, with a quick recovery and favorable outcome [51]. Among North American patients, both fumarates and natalizumab treatments were each associated with decreased odds of ICU admission and/or ventilation need [25]. In these patients, antibodies against SARS-CoV-2 were detected soon after the infection, and repeated negative results for Polymerase Chain Reaction (PCR) for SARS-CoV-2 after symptom resolution reflect a low replication rate. Nonetheless, twenty (4.3%) of the patients with COVID-19 and natalizumab described in the literature had a fatal outcome, mainly patients with other comorbidities such as coronary artery disease, hypertension and obesity [11,18,21,25,52]. Whether altered dynamics of peripheral immune cells in patients on natalizumab could worsen the «cytokine storm» syndrome associated with severe COVID-19, the contribution of natalizumab to these poor outcomes remains unknown [51]. So far, it appears safe to start or proceed with natalizumab during the pandemic, including during acute infection, despite a somewhat hypothetical increased risk of SARS-CoV-2 encephalitis, due to reduced CNS immunological surveillance. The adoption of an extended interval dosing (EID), with perfusions every 6-weeks, proved to be an equally effective alternative at a reduced risk of exposure. Borriello G et al. described a patient who was administered natalizumab in the EID immediately after recovering from COVID-19, without any worsening or new symptoms reported [50], supporting the notion that discontinuing or delaying the treatment further than this period, even in patients recently recovered from COVID-19, is unnecessary.

Anti-CD20 monoclonal antibodies act by promoting a protracted B lymphopenia and an overall 25% reduction in total lymphocyte count. Long term treatment and memory cell pool depletion also portend a higher risk of hypogammaglobulinemia and a consequent greater risk of infection relatively to other DMTs [7,18,26]. Several reports of COVID-19 in patients receiving anti-CD20 monoclonal antibodies have been published. Critical evidence also was derived from pharmacovigilance studies, especially the Roche^®^ (Basel, Switzerland)/Genentech^®^ (San Francisco, CA, USA) global safety databases, with over 160,000 MS patients worldwide. We identified 1215 cases of confirmed (*n* = 1042) or suspected (*n* = 172) COVID-19 in MS patients treated with ocrelizumab [8,11,13,15,16,18,19,22,23,24,25,29,35,53,54,55,56,57,58,59,60,61,62,63]. Thirty-five (2.8%) of these patients had a fatal outcome. The majority were patients with progressive MS and high EDSS, with other comorbidities such as prostatic cancer [11], chronic obstructive pulmonary disease (COPD) [20], obesity or hyperpolymenorrhea [18,56,58]. A 27-year-old female patient on ocrelizumab also succumbed to the disease but she had polycystic ovaries, depression, anxiety and was prescribed numerous concomitant medications [58]. In a North American real-world series, patients on ocrelizumab showed increased odds of hospitalization, ICU admission (40.3% of the patients) and fatal outcome (22.9% of the patients) [25]. It is worth to note that our results are likely to reflect some overlap between isolated case reports and those reported in the Roche^®^ (Basel, Switzerland) safety database [55]. Regarding rituximab group, we identified 236 patients with MS who had been infected with SARS-CoV-2 [7,8,11,12,15,22,23,24,25,54,59,64,65,66,67,68]. Among these, eight (3.4%) patients died. Again, comorbidities (Hodgkin lymphoma, venous thrombosis) [11] and patient characteristics (SPMS, EDSS higher than 6.5) [12,15,25], may justify an expected worse outcome. In the Italian series, anti-CD20 therapy (Ocrelizumab or Rituximab) was significantly associated with an increased risk of severe COVID-19 course, even after adjusting for age, sex and progressive MS [18]. The same was replicated in Iranian patients [7,15], despite the absence of an increased odds of hospitalization in this series. In a retrospective cohort study using electronic health records in California, rituximab-treated MS patients (*n* = 24) were more likely be hospitalized (*n* = 8, 33.3%), but not die (*n* = 0) from COVID-19 than a non-MS population (5.8% and 1.4%, respectively) [67]. Similarly, rituximab-patients seem to have 4.5-fold increased odds of hospitalization and possibly worse outcomes (albeit the last was not statistically significant) [25]. Notably, the increased risk of a moderate to severe COVID-19 course occurred shortly after the most recent rituximab treatment (median 2.5 months, range 2 days to 4.5 months) and was more likely to occur for the 1000 mg dosage at last infusion or a higher cumulative dose. Since the anti-CD20 therapies act primarily on B cells, they affect not only the early immune response (given their role as antigen-presenting cells to T cells) but also late and prolonged immune responses, increasing the susceptibility to re-infection [59,68]. Many case reports describe attenuated humoral responses after infection in both ocrelizumab [22,60,62,63] and rituximab-treated patients, with possible vaccination repercussions, as it will be further discussed [65,66]. Importantly, the number of B cells neither the antibody titer seem to impact the prognosis, as the absence of CD19+ B cells occurs in the majority of the patients, yet they had different disease courses, and asymptomatic carriers still developed antibodies. This is reinforced by the fact the quick recovery of two cases of COVID-19 with pneumonia and lymphopenia in patients with X-linked agammaglobulinemia [69]. Since some patients with negative serology came off rituximab treatment before ocrelizumab, perhaps the use of both therapies or longer treatment period with anti-CD20 therapies might be anticipated as detrimental for antibody formation [59]. Given it all, starting ocrelizumab during the pandemic should be carefully weighed against the risks and the prevision of vaccination starting around the globe for MS patients (namely for progressive MS, for which ocrelizumab is the single approved drug). For those on treatment, an extension of the perfusion interval based on anti-CD19 count is highly recommended, while the timing of treatment after SARS-CoV-2 infection requires further investigation and individual consideration to reduce the risk of reactivation. Further, lower doses should be considered when possible and MS patients on anti-CD20 therapies should be counseled to take extra precautions in the 5 months following each infusion. More recently, Flores-Gonzalez [70] reported on a MS patient on subcutaneous ofatumumab (approved in 2020 by the US Food and Drug Administration as once-monthly subcutaneous injection waiting approval by the EMA) who remained asymptomatic during SARS-CoV-2 infection and still developed neutralizing IgM and IgG antibodies three months after the infection, despite B-cell depletion. Additionally, data derived from the open-label extension study ALITHIOS or the post-marketing setting, recently presented at the Annual Meeting of Neurology, reported on 12 cases (11 non-serious; one serious hospitalized for bilateral pneumonia) of laboratory-confirmed SARS-CoV-2 infection in patients with MS treated with ofatumumab [71]. Further research will be necessary to evaluate the humoral response of MS patients on these brand-new anti-CD20 therapies to SARS-CoV-2 infection and COVID-19 vaccines.

Alemtuzumab is a humanized monoclonal antibody against CD52, a receptor located on mature lymphocytes surface. It causes generalized lymphopenia, mainly of the major circulating T lymphocytes (CD3+, CD4+ and CD8+), similarly to an immune reset, followed by immune reconstitution. It usually promotes sustained disease remission after two or three treatment cycles, removing the need for long-term treatments. By acting on CD52, it also targets the innate immune system (by activation of pro-apoptotic pathways on macrophages and dendritic cells), at a risk of neutropenia and even pancytopenia, demanding frequent follow-up [72]. The potential negative side effects are usually maximal during the first six months after infusion, when the risk of lymphopenia is higher [26]. Patients on alemtuzumab may possibly face an increased susceptibility to SARS-CoV-2 infection and re-infection, as they are deprived of both early and late immune responses. We identified 58 alemtuzumab-treated MS patients with COVID-19 (confirmed *n* = 35) [8,13,16,17,18,19,20,21,22,23,24,25,31,73,74,75,76,77,78], one of them with a fatal outcome [25]. Some cases occurred shortly after the second cycle of alemtuzumab (between one week and 2-months) and the patients had moderate to severe lymphopenia (approximately 85% from lower limit of normal), despite a mild and self-limiting presentation, at no need for hospitalization. B lymphocytes and NK cells were usually between normal limits [73,75]. These cases show that during the immune reconstitution, after two or three treatment cycles and normal lymphocyte count, an increased susceptibility to SARS-CoV-2 infection is highly unexpected. Nonetheless, these patients might not mount an effective humoral response, at risk of re-infection and decreased vaccination response [22]. As so, starting alemtuzumab while the pandemic is still expanding is not amendable if other treatment options are allowed to be considered.

Cladribine is a purine nucleoside analog that interferes with cellular metabolism and inhibits DNA synthesis and repair, causing significant myelosuppression by cell apoptosis, especially in lymphocytes (mainly CD4+ and CD8+ T cells, but also B cells) [26,42]. Accordingly, during the depletion phase (usually the first 6 months) transient lymphopenia (often mild to moderate) may subside, as well as minor effects on innate immune cells such as neutrophils, monocytes and NK cells [42]. Patients on cladribine are at risk of severe infections, including SARS-CoV-2. Given the effects on early and late immunity, a risk of re-infection is not trivial. Like alemtuzumab, immune reconstitution occurs after the second cycle, avoiding the need for long-term therapy. We identified 335 patients who had COVID-19 under cladribine (confirmed cases *n* = 195) [8,17,18,19,20,23,25,31,79,80,81,82,83,84,85]. A recent update on Merck^®^ (Darmstadt, Germany) KGaA Global Patient Safety pharmacovigilance database through 15 January 2021 (272 reported cases), described the first COVID-19 dead in a MS patient treated with cladribine [83]. On the same series, 40 patients (15%) experienced serious COVID-19, while 133 (51%) patients had a complete recovery. The median time from the most recent preceding cladribine treatment course to onset of COVID-19 was 162 days. We are aware that the cases reported in the database may overlap with other published reports that we here present, thus it is difficult to ascertain the precise number of cladribine patients affected by COVID-19. Regarding immune responses and antibody formation to SARS-CoV-2, an adequate immune response with detectable antibodies three months after infection was reported in two MS patients on cladribine, despite the low lymphocyte levels [80,81]. On the contrary, a 32-years-old female MS patient with self-limiting COVID-19 disease, developing approximately four weeks after treatment course, showed negative IgM and IgG anti- SARS-CoV-2 antibodies one month after the infection [85], similarly to a case described by Preziosa et al. [84]. The findings collected so far do not support a greater risk of severe outcomes in cladribine-treated patients, even when infection occurs in the first weeks after treatment. Postponing the treatment until the recovery of COVID-19 is highly advisable [85]. Comparatively to other DMTs and to the general population, no severe outcomes (no pneumonia, hospitalization, ICU, or death) were observed in patients treated with both cladribine or alemtuzumab [18]. In summary, growing biological and clinical evidence show that most MS treatments, especially those with limited longstanding effects on the innate immune system or CD8 T cell responses, have a negligible influence on COVID-19 incidence and outcomes. Conversely, those potentially causing severe lymphopenia and hypogammaglobulinemia might portend a higher risk [22].

**Table 1 vaccines-09-00773-t001:** Disease modifying therapies in Multiple Sclerosis patients with COVID-19.

Drug	COVID-19 Cases		Death See text for details	Total (%)	References
	Confirmed	Suspected			
Beta-interferon	144	74	4	218 (4.9)	[7,8,9,10,11,12,13,14,15,16,17,18,19,20,21,22,23,24,25]
Glatiramer acetate	196	73	4	269 (6.1)	[8,10,11,15,16,18,19,20,21,22,23,24,25,29,30,31]
Dimethyl fumarate	408	195	19	603 (13.7)	[8,11,12,13,14,15,16,17,18,19,20,21,22,23,24,25,29,31,35,40,41]
Teriflunomide	202	49	4	251 (5.7)	[7,8,10,13,15,16,17,18,19,20,21,22,23,24,25,31,32,33,34,35,36,37,38,39]
Dimethyl fumarate/teriflunomide ^§^	-	108	-	108 (2.4)	[30,31]
Fingolimod	268	146	1	414 (9.4)	[7,8,10,11,14,15,16,17,18,19,20,22,23,24,25,30,31,35,37,43,44,45,46,47,48,49]
Siponimod	19	-	2	19 (0.4)	[11,25]
Ponesimod	-	1	-	1 (0)	[18]
Non-specified S1P ^1^ modulator	29	-	-	29 (0.7)	[21]
Natalizumab	325	136	20	461 (10.4)	[8,11,13,14,15,16,18,19,20,21,22,23,24,25,30,31,35,50,51,52]
Alemtuzumab	35	23	1	58 (1.3)	[8,13,16,17,18,19,20,21,22,23,24,25,31,72,73,74,75,76,77]
Cladribine	195	140	1	335 (7.6)	[8,17,18,19,20,23,25,31,78,79,80,81,82,83,84]
Alemtuzumab/cladribine ^§^	-	15	-	15 (0.3)	[30]
Ocrelizumab	1042	173	35	1215 (27.5)	[8,11,13,15,16,18,19,22,23,24,25,29,35,53,54,55,56,57,58,59,60,61,62,63]
Rituximab	211	25	8	236 (5.3)	[7,8,11,12,15,22,23,24,25,54,59,64,65,66,67,68]
Ofatumumab	13	-	-	13 (0.3)	[70,71]
Non-specified anti-CD20	123	49	-	172 (3.9)	[21,30,31]
Total	3210	1207	99	4417 (100)	

^1^ S1P-sphingosine-1-phosphate; ^§^ Two studies [30,31] reported patients suspected to have COVID-19 on either Dimethyl fumarate/teriflunomide (*n* = 108) or Alemtuzumab/cladribine (*n* = 15) (non- specified).

### COVID-19 Vaccine in MS Patients on DMTs

The COVID-19 vaccination has been heralded as a key step to overcome this pandemic. As the virus uses its outer spike protein (S protein) to bind to angiotensin-converting enzyme-related carboxypeptidase 2 (ACE2) on the host cell surface [86], many vaccines use this protein as their target antigen [87,88]. Despite their distinct mechanisms of action, they all intend to mimic the natural process of infection. In December 2020, the lipid nanoparticle-formulated Pfizer-BioNTech^®^ (New York, USA/ Mainz, Germany) (BNT162b2) COVID-19 vaccine [89] was the first to receive conditional emergency approval worldwide, followed by the mRNA-1273 vaccine from Moderna^®^ (Cambridge, MA, USA) [90] in January 2021. Both feature viral genetic material and are based on nucleoside-modified mRNA vectors encoding the spike glycoprotein of SARS-CoV-2. These vaccines have proven an efficacy superior to 90% in preventing severe and mild forms of COVID-19, independently of age, race and certain comorbidities including asthma, COPD, diabetes, hypertension and obesity (BMI ≥ 30 kg/m^2^) [89,90]. COVID-19 Vaccine AstraZeneca^®^ (Cambridge, UK) (Vaxzevria) is an adenoviral vector-based vaccine in which the DNA encoding the coronavirus spike protein antigen is cloned into a viral vector that lacks the ability to reproduce and cause disease itself. The vaccine has demonstrated an efficacy ranging from 60–94% at protecting people from the extremely serious risks of COVID-19, including death, hospitalization and severe disease in clinical trials, including in aged people over 65 years old [91,92]. The most common side effects with AstraZeneca^®^ (Cambridge, UK) vaccine were usually mild or moderate and typically short-limited. Similarly, another adenovirus-based vaccine from Janssen^®^ (Beerse, Belgium) led to 67% reduction in the number of symptomatic COVID-19 cases weeks after immunization, also with mild or moderate side effects which mostly consisted of pain at injection site, headache, tiredness, muscle pain and nausea. New very rare side effects have been recognized such as embolic and thrombotic events with a focus on thrombosis in combination with thrombocytopenia for both the Janssen^®^ (Beerse, Belgium) and Vaxzevria COVID-19 vaccines. With any of these vaccines, the goal is to activate the immune system, namely B and T cells to produce neutralizing antibodies (equivalent to the titer found in convalescent patients) and prevent a future infection. While vaccines from Pfizer-BioNTech^®^ (New York, NY, USA/Mainz, Germany) and Moderna^®^ are administered intramuscularly in two administrations 21 days or 28 days apart, respectively, AstraZeneca^®^ (Cambridge, UK) vaccine requires two intramuscular injections administered 4–12 weeks apart. Other new generation vaccines are under study and include recombinant protein vaccine, bacterial vector-based vaccine, plasmid DNA vaccine and trained immunity-based vaccine.

While vaccine supplies are still limited, priority is being given to healthcare workers and others at higher risk of exposure or complications such as older and immunosuppressed patients. As live and attenuated viruses’ vaccines are contraindicated in MS patients, these new DNA-RNA vaccines are well welcomed in patients on DMTs [93]. Yet, common concerns of the population and medical community are vaccine safety and effectiveness. Recent studies have focused on the willingness of MS patients to get the COVID-19 vaccine. 80.9% of European MS patients were willing to receive the vaccine, however 54.1% would prefer to postpone vaccination until they seek advice of their physician. Interestingly, older patients and those with comorbidities were the ones with the biggest interest in getting the protection of the vaccine [94]. In the US, the acceptability was slightly inferior, but 66% of the participants were still willing to get a COVID-19 vaccine. The information sources most highly trusted were healthcare providers and the National MS Society, confirming the importance of review studies and expert recommendations [95]. Several studies have shown that there is no difference in vaccine responses between MS patients and healthy individuals [96] but, given the variable immune responses and the absence of clinical trials in this population, the safety and efficacy of approved SARS-CoV-2 vaccines in MS patients is still unknown to us, [37,62,63,81,93,96]. A recent study conducted in Israel, a country where vaccination has been a keen priority, evaluated the adverse event profile and immediate risk of acute relapses in 555 MS patients who received the COVID-19 BNT162b2 vaccine [97]. No events of anaphylaxis or life-threatening responses were registered. Between the two doses, three patients (0.5%) were infected by SARS-CoV-2, all being asymptomatic or with mild disease. A pseudo-relapse with flu-like symptomatology was reported in 2% and 4.8% of patients after the first and second vaccination doses, respectively. Acute relapses shortly after vaccination corresponded to the expected relapse rate and so the vaccine was not associated with increased disease activity.

Several questions regarding the timing of the vaccine and drug administration need to be addressed [96]. Despite one study found a negative impact of GA on immune response to Influenza vaccine, it was not replicated in subsequent studies, and so it seems safe to administer a vaccine against SARS-CoV-2 in MS patients treated with both IFN or GA [96,98] Similarly, based on previous observations from Influenza vaccine responses in MS patients treated with teriflunomide [99] and confirmed seroconversion after COVID-19 [37], a successful vaccination can be anticipated in patients treated with teriflunomide [38]. Regarding fumarates and natalizumab, previous response to other vaccines suggest that patients can mount an adequate seroprotective response to inactivated or protein-based vaccines, probably because they retain functional T and B cells and stable serum immunoglobulins [96,100]. Given its mechanism of action and reduced immunoglobulin responses to SARS-CoV-2, concerns might be anticipated about the impact of fingolimod on vaccination [37]. As so, serology status should be tested after vaccination, to ensure serological protection. Nonetheless, MS patients with lymphopenia still mounted anti-SARS-CoV-2 humoral responses suggesting that COVID-19 vaccination might hold out an attenuated but still protective response in this vulnerable population. For IFN, GA, DMF, teriflunomide and fingolimod, therapeutic withdrawal as a mean to increase vaccination efficacy is not encouraged.

On the other hand, for anti-CD20 monoclonal antibodies, a blunted seroconversion might be anticipated in the 6–10 months after the last infusion, allowing for a time-window vaccination that might be programmed based on the kinetics of repopulation of the naive cell pool with immature B cells. Memory B cell repletion can take up to 18 months after discontinuation of ocrelizumab, and up to 11–12 months for rituximab and ofatumumab [93]. Additionally, the presence of worsening hypogammaglobinemia with repeated infusions may contribute to suboptimal serological responses. In a patient series, less than 20% of MS patients treated with ocrelizumab generated an antibody response when naturally infected by COVID-19 [22]. Further, attenuated humoral responses to tetanus, seasonal flu and pneumococcus vaccines were detected in B-cell-depleted ocrelizumab MS patients in the VELOCE study (a randomized, open-label, Phase III trial) [101]. Recently, real-world data on COVID-19 vaccination and ocrelizumab were revealed. While a 64-year-old female patient, treated with ocrelizumab and vaccinated 3 months after the last infusion, was able to produce a protective antibody response against COVID-19, another 46-year-old female patient treated with ocrelizumab, who received the vaccine 2 months after the last infusion, did not produce a serological response [102]. Both had similar low CD19 counts. The same was observed in a patient who received the second dose of COVID-19 vaccination nine days after last ocrelizumab infusion [103]. Further, vaccination failure was described for a 52-year-old male MS patient on ocrelizumab who developed his first symptoms of SARS-CoV-2 infection 19 days after receiving the last dose of the COVID-19 vaccine [104]. The role of genetic discordance of the different SARS-CoV-2 strains to these observations remains a significant but unanswered question [105]. As for COVID-19 outcomes, the dose and treatment duration may have a significant impact on vaccination efficacy, especially in ocrelizumab patients previously treated with rituximab [93]. Importantly, as ocrelizumab does not appear to modify pre-existing humoral immunity, MS patients might safely resume ocrelizumab 4–6 weeks after receiving SARS-CoV-2 vaccine. For those already taking anti-CD20 therapies, administered on a 6-month interval schedule, vaccination should be deferred toward the end of the cycle, at least 12 weeks after the last drug dose [101]. For the newer monthly administered subcutaneous B cell–depleting therapy ofatumumab, vaccination might be delivered toward the end of the monthly cycle, and the next two ofatumumab doses skipped to allow for the booster vaccine to take effect.

Finally, for the two long-term immune-depleters, cladribine allows CD19 naive B cells to recover in approximately 30 weeks [106], while for alemtuzumab there is usually a rapid repopulation of naive B cells, despite a marked memory B cells depletion. For both DMTs, a reduction in cellular and humoral responses to COVID-19 vaccines is expected during maximal lymphocyte depletion, so that vaccination timeline is dependent on immune reconstitution. In clinical trials, patients under cladribine mounted immune response to influenza vaccine four weeks after vaccination, without additional adverse events [106], while a blunted antibody response to vaccination has been observed in the first six months after alemtuzumab treatment [107]. For alemtuzumab, delaying vaccination for at least six months after the last treatment cycle and adjust the second cycle to ensure an optimal vaccination response is highly advisable [73,75,93,96]. Similarly, for cladribine we recommend a three-month gap after the treatment cycle until vaccination (or until the recovery of lymphocyte count) [108]. In line with it, real-world data show that the first two patients under cladribine receiving the adenoviral vector-based vaccine (AstraZeneca^®^ (Cambridge, UK) and the Pfizer-BioNTech^®^ (New York, NY, USA/ Mainz, Germany) mRNA vaccine three months after the second cycle of treatment exhibited a protective antibody response despite an incomplete reconstitution of the absolute values of circulating lymphocytes [102]. As for other DMTs, checking serological status after vaccination is encouraged, especially during the lymphocyte depletion phase. Additional booster vaccinations for patients demonstrating insufficient responses might be amenable.

Apart from B cell responses, the role of T cell-induced responses (as it occurs for herpes zoster vaccine) remains to be studied but an ongoing pioneer prospective study analyzing this aspect confirms that ocrelizumab-treated patients who get COVID-19 have good robust T-cell responses, suggesting persistent T-cell immune memory to SARS-CoV-2 up to 10 months following infection, even in B-cell depleted MS patients [109]. The question of whether patients fail to respond to the vaccine at both an antibody and T-cell level remains to be solved. Further, for patients who have received high-dose steroids in the last month, it is likely favorable to postpone vaccination [18], despite the lack of data on vaccine efficacy in people treated with high doses of corticosteroids (≥20 mg prednisone equivalents for ≥14 days) [96]. Finally, given their more advanced age and greater disability, patients with progressive MS face an increased risk of severe COVID-19 and frequently do not receive DMTs (unless in early and active phases). Vaccination benefits seem to clearly outweigh any potential risk associated with vaccine administration in this group.

## 4. Discussion

Since the beginning of the pandemic, our understanding of the correlation between MS, DMTs and COVID-19 outcomes has been further consolidated by biological and clinical evidence derived from published case reports, small cohorts and pharmacovigilance registries (especially Biogen^®^ (Cambridge, MA, USA) global safety database, Roche^®^ (Basel, Switzerland) and Merck^®^ (Darmstadt, Germany) databases as well as real-world COVID-19 registries such as MuSC-19, COVISEP, and the MS Global Data Sharing Alliance) [21]. Currently, available evidence does not support the notion that MS patients per se are more prone to severe COVID-19 [8,18,25]. Instead, the risk of severe disease is notably influenced by age and comorbidities, such as diabetes, obesity (body mass index ≥ 30), comorbidity score ≥ 1, COPD, chronic kidney disease and Black/African ancestry, as described for the general population [8,18,21]. EDSS was also identified as a risk factor with a significant impact on COVID-19 severity [18]. A protective effect of DMTs in patient groups without these comorbidities remains to be established. The focus is now turning away from risks associated with SARS-CoV-2 to guarantee of immune protection in those uninfected MS patients. Currently, there are no approved guidelines for the administration of COVID-19 vaccines in MS patients treated with DMTs [108]. To date, available evidence shows that COVID-19 vaccines are effective in reducing the risk of SARS-CoV-2 infection also in MS patients, without triggering disease exacerbation, MS relapses, or increased adverse effects. They do not seem to interfere with DMTs effectiveness as well. In this study, we reviewed the available evidence on the immune response duration with implications in the risk of re-infection, which are determinant aspects that still need to be sought and clarified. Almost all classes of DMTs might reduce vaccine effectivity by immune system suppression or modulation, except for IFN. This effect is probably more remarkable in patients on anti-CD20 therapies, especially in the following months after infusion, when B cell depletion is more significant [93,101]. As a rule of thumb, untreated MS patients should be promptly vaccinated (especially if they are soon to be started on fingolimod, alemtuzumab, ocrelizumab and oral cladribine therapy), as well as those treated with IFN, natalizumab or GA. For fingolimod, DMF, teriflunomide, anti-CD20 therapies, alemtuzumab and cladribine, pre-vaccination lymphocyte count is advisable, and vaccination recommended without the need of stopping the immunomodulatory treatment [110], as long as vaccination is timed according to individual patient characteristics and DMT specific mechanism of action [102,108]. Nonetheless, the suggested scheduling might not always be possible for every patient. Cost-benefit of treatment strategies, including treatment changes or first prescriptions, and the need to vaccinate in a reasonable time during COVID-19 pandemic might need to be individually tailored (Table 2) [18].

## 5. Conclusions

Our study is a contemporary review of COVID-19 data in MS patients throughout April 2021, when vaccination campaigns have started across the globe. The data presented in this study certainly have some limitations. First, it is not possible to deduce the exact risk of COVID-19 associated with individual DMTs as we can only rely on published cases and there might be a tendency to over-representation of some drug classes depending on pharmacovigilance registries. Second, severity of COVID-19 and its complications is hardly comparable among the cases described in the literature because of the omission of some important details in many case series/reports (e.g., hospital discharge, PCR negativity, symptom resolution, return to work or usual activities). Third, conclusions derived from case reports and case series are limited by small sample size and not free from multiple possible sources of bias (namely selection bias and report bias). Finally, despite the clinical course and outcomes of COVID-19 in MS patients seem to be in line with the general population for most DMTs, we cannot rule out that these positive outcomes might not be due to a specific immunomodulator effect, since most COVID-19 patients recover spontaneously, without directed treatment. Finally, despite some data on MS patients and COVID-19 vaccination are starting to emerge, further studies are needed to help guide treatment strategies and optimize success with vaccination protocols, while minimizing treatment interruption risks. Continuous recordings on pharmacovigilance and MS registries will be determinant for a generalization of the efficacy and safety of the COVID-19 vaccine in this particular population.

## Figures and Tables

**Figure 1 vaccines-09-00773-f001:**
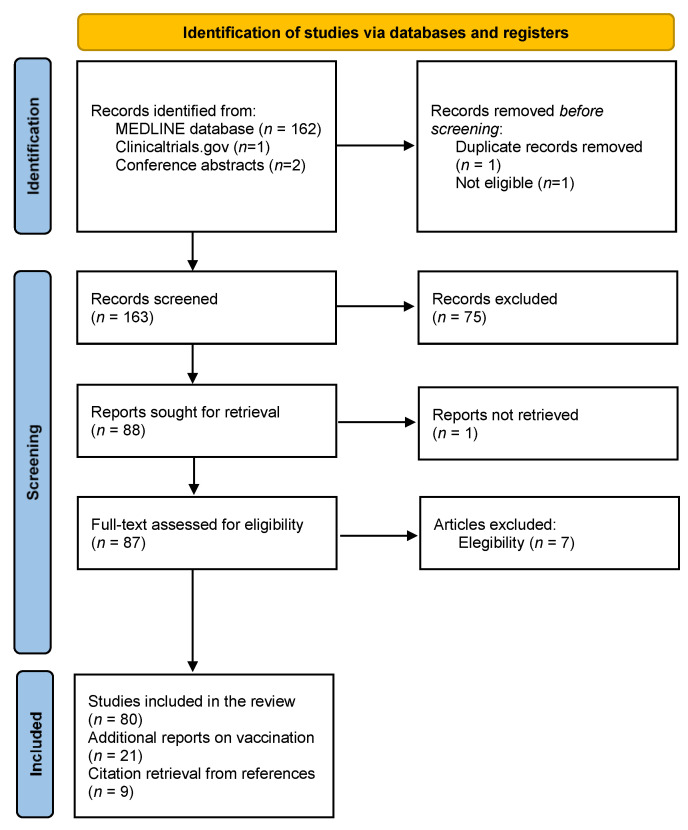
Flow-chart representing study selection according to PRISMA criteria.

**Table 2 vaccines-09-00773-t002:** Impact of Multiple Sclerosis disease modifying therapies on vaccination and expert recommendations.

Drug	Impact on Vaccination Response	Recommendation
Interferons	No impact	Similar to other vaccines
Glatiramer acetate	Some studies have suggested a blunted humoral response to Influenza vaccine. No data for other vaccines.	If possible, vaccination must be administered previously to first drug administration
Terifluonomide	Possibly no impact	If possible, vaccination must be administered previously to first drug administration
Dymethil fumarate	Response to toxoid, conjugate and polysaccharide vaccines was not affected	If possible, vaccination must be administered previously to first drug administration, due to lymphopenia risk
S1P modulators ^1^	Reduced response to inactivated, toxoid and polysaccharide vaccines with fingolimod Slightly blunted response to Influenza vaccine with Siponimod	If possible, vaccination must be administered previously to first drug administration
Cladribine	No specific studies but MS ^2^ patients under cladribine have mounted immune response to influenza vaccine after four weeks from vaccination, without additional adverse events. COVID-19 vaccine three months after the second cycle of treatment promoted a protective antibody response despite an incomplete immune reconstitution.	A three-month gap after the treatment cycle until vaccination is recommended (or until the recovery of lymphocyte count)
Natalizumab	Possibly no impact	If possible, vaccination must be administered previously to first drug administration
Anti-CD20	Attenuated humoral responses to tetanus, seasonal flu, pneumococcus and SARS-CoV-2 vaccines were observed	Ocrelizumab/rituximab: vaccination should be deferred toward the end of the cycle (12 weeks or more after the last drug dose) and the next drug dose administered at least 4–6 weeks after completing vaccination. Ofatumumab: vaccination might be delivered toward the end of the monthly cycle and the next two ofatumumab doses skipped.
Alemtuzumab	Blunted immune response until six months after last treatment cycle, but retained after that period	Vaccination should be delayed for at least six months after the last treatment cycle and the second cycle adjusted to ensure an optimal vaccination response.
All	-	Live vaccines are generally contraindicated. Pre-vaccination lymphocyte count is advised. Treatment withdrawal to promote vaccination response is not recommended. Post-vaccination serology status checking is encouraged.

^1^ S1P—sphingosine-1-phosphate; ^2^ MS—Multiple Sclerosis.

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
