# Peer review of "Multiple Sclerosis, Disease-Modifying Therapies and COVID-19: A Systematic Review on Immune Response and Vaccination Recommendations"

_vaccines, 2021, doi:10.3390/vaccines9070773_

Round 1

Reviewer 1 Report

The Authors made a systematic review on immune response and COVID-19 vaccination recommendations in pwMS taking disease-modifying therapies. The review is interesting and complete. I have only 2-3 questions7suggestions.

Table 1 I did not understand why there is a column with "Dimethylfumarate/teriflunomide" and "Alemtuzumab/cladribine". Can you clarify it? maybe in the legend.

As regards the time of administration of the vaccine AFTER ocrelizumab the literature is currently very discordant.

A time interval of six months is certainly very safe, but in some cases not practicable.

At the present time, the need to vaccinate patients in a reasonable time could outweigh the risk of not having a complete immune response.

there is evidence, although limited, that vaccination at least three months after Ocrelizumab administration may be safe and effective. Could you comment on it?

line 121 there is a typo.

Author Response

The authors thank the reviewer for the proposed questions/suggestions that certainly enrich and clarify the messages we want to convey.

Commentary 1: "Table 1 I did not understand why there is a column with "Dimethylfumarate/teriflunomide" and "Alemtuzumab/cladribine". Can you clarify it? maybe in the legend."

> Response: The following sentence was added to the legend: "Two studies (30,31) reported 108 and 15 patients suspected to have COVID-19 on either Dimethyl fumarate/teriflunomide or Alemtuzumab/cladribine (not further specified), respectively."

Commentary 2: "As regards the time of administration of the vaccine AFTER ocrelizumab the literature is currently very discordant. A time interval of six months is certainly very safe, but in some cases not practicable. At the present time, the need to vaccinate patients in a reasonable time could outweigh the risk of not having a complete immune response. there is evidence, although limited, that vaccination at least three months after Ocrelizumab administration may be safe and effective. Could you comment on it?"

> Response: In fact, despite six-month interval would be the ideal time to ensure a complete immune response, evidence from the VELOCE study seems to guarantee a still protective immune response when the vaccines are administered three months/12 weeks after anti-CD20 administration. We clarified this aspect in the text (please check lines 443-448 and table 2 of the manuscript).

Commentary 3: "line 121 there is a typo."

> Thank you. It was corrected (changes highlighted in the document).

Reviewer 2 Report

STRENGHT risks of COVID-19 in patients with Multiple Sclerosis (MS) and their immune reactions according to COVID vaccine responses is reviewed. COVID-19 course analized in MS patients during April 2021, when vaccination campaigns started is presented and outcomes in patients receiving disease modifying therapies was conducted according to the Preferred Reporting Items for Systematic Reviews and Meta-Analyses.

Data on SARS-CoV-2 vaccines is used and reccomendation made, the available evidence on the Duration of the immune response as well as the risk of re-infection, are aspects covered.

it is stated that is not possible to deduce the exact risk of COVID-19 associated with individual DMTs as we only rely on published cases and there might be a tendency to over-representation of some drug classes depending on pharmacovigilance registries so more data on pharmacovigilance will also be determinant to allow for a generalization of the efficacy and safety of the COVID-19 vaccine.

Weakness: please check phrases in english to become more undestandable and check carefully the language by a native speaker

Author Response

The authors thank the reviewer for this nice and detailed overview of our paper.

Regarding the pointed weakness concerning English language, we carefully reviewed the manuscript and made changes in longer phrases to make them more readable and undestandable.

Reviewer 3 Report

In this MS, Authors Verónica Cabreira and collaborators, propose a systematic review on multiple sclerosis (MS) and disease-modifying therapies (DMTs) in COVID 19 pandemic situation and vaccine-associated context.

As far as vaccine response against COVID 19 for MS patients under DMTs is concerned, this MS deals about a subject of great interest, especially in the current pandemic situation.

The MS is 18 pages long, contains one figure, two tables and cites 110 quite recent publications.

Introduction correctly states the problematic and rationale of the MS.

Material and Methods. PRISMA based recommendations to perform the systematic review seem to be very adequately followed.

I appreciated Authors specified action mode of each DMTs before analysing their results in term of impact of COVID 19 infection. Authors also nicely described the action mode of each vaccine before analysing effect in MS patients under DMTs.

That said I have minor points to address:

Do Authors have retrieve data on COVID 19 impact on MS patients having no DMTs?

Middle of page 5. The sentence starting with “The drug a dual antiviral…” is missing a verb, pls check.

Author Response

The authors thank the reviewer for this very nice and detailed overview of our paper and the comments/suggestions that certainly enrich the content we here present. We now provide the answers for the minor points that required our attention.

Commentary 1: "Do Authors have retrieve data on COVID 19 impact on MS patients having no DMTs?"

> R: Despite falling outside the scope of our paper (and search strategy), so that data on COVID-19 severity and outcomes in non-treated MS patients is not presented by us, we briefly mention this population in two occasions (under the vaccination topic), as follows:

line 482: "Finally, given their more advanced age and greater disability, patients with progressive MS face an increased risk of severe COVID-19 and frequently do not receive DMTs (unless in early and active phases), and so the vaccination benefits clearly outweigh any potential risk associated with vaccine administration in this group."

line 513: "As a rule of thumb untreated MS patients should be promptly vaccinated (especially if they are soon to be started on fingolimod, alemtuzumab, ocrelizumab and oral cladribine therapy), as well as those treated with IFN, natalizumab or GA."

Commentary 2: "Middle of page 5. The sentence starting with “The drug a dual antiviral…” is missing a verb, pls check."

>R: Thank you. The changes were made accordingly.